# Retrieval-Free Instruction Selection for Instruction-Tuned Embedding Models via Uncentered Spectral Entropy

## Abstract

Instruction-tuned embedding models expose a consequential deployment variable: the query-side instruction. Choosing that instruction usually requires repeated retrieval evaluation with corpus access and judged queries, precisely when such infrastructure is least available: cold-start domains, API-only embedding services, and large candidate pools that must be screened before a stable retrieval stack exists. We study this earlier decision point and ask whether candidate instructions can be ranked before retrieval evaluation is practical. Our central hypothesis is geometric: better instructions induce a broader, less collapsed representation geometry on a small unlabeled set of query-like proxy texts. Based on this view, we propose *Instruction Performance Prediction* (IPP), a retrieval-free, label-free screening method that scores each instruction by the normalized spectral entropy of the second-moment matrix of its proxy embeddings. Across 16 embedding models, 17 retrieval datasets, and all 272 model–dataset pairs (full $16 \times 17$ coverage), IPP attains median oriented Spearman $\rho = 0.806$ and median regret@1 of 0.004 NDCG@10 points on a 104-instruction pool. The same evidence also identifies a clear operating boundary: when candidate instructions produce little downstream variation, the ranking problem becomes weakly separable, so geometric screening should hand off to direct retrieval evaluation rather than be over-interpreted.

## 1 Introduction

As information retrieval becomes a core subsystem of larger AI systems, from retrieval-augmented generation to autonomous agents, the interface exposed by the retriever increasingly shapes end-to-end behavior (Lewis et al., 2020). A common interface in this setting is the instruction-tuned embedder, where the query is paired with a short natural-language instruction such as `query:` or `search_query:` before encoding (Su et al., 2023; Wang et al., 2024a;b; Chen et al., 2024; Zhang et al., 2025; Hu et al., 2025). These instructions behave like human-readable prefixes: they can reorganize the embedding space without changing the model weights. Instruction choice is therefore a system decision, not benchmark decoration.

That decision is consequential. On FiQA in BEIR with E5-base-v2, changing only the query-side instruction creates a 16.7-point NDCG@10 gap between the best and worst candidate in a 104-instruction pool, even though the checkpoint, document index, and ranking pipeline remain fixed. Yet selecting an instruction by brute force requires repeated retrieval evaluation over queries, a corpus, and relevance judgments. This is most painful exactly where early screening would be most valuable: cold-start domains without mature qrels, API-only embedding services that restrict corpus-side experimentation, and large candidate pools that are too expensive to exhaustively test at the outset.

Existing query performance prediction (QPP) methods address a related but later problem. They estimate the difficulty of a *query* once a retriever, a corpus, and retrieval-time signals already exist (Carmel & Yom-Tov, 2010; Datta et al., 2025). Our setting is different in both unit and stage: the object to be ranked is an *instruction*, and the ranking must happen *before* instruction-by-instruction retrieval evaluation is practical. The question is therefore not whether a query is hard after retrieval, but whether an instruction is promising before retrieval evaluation is feasible.

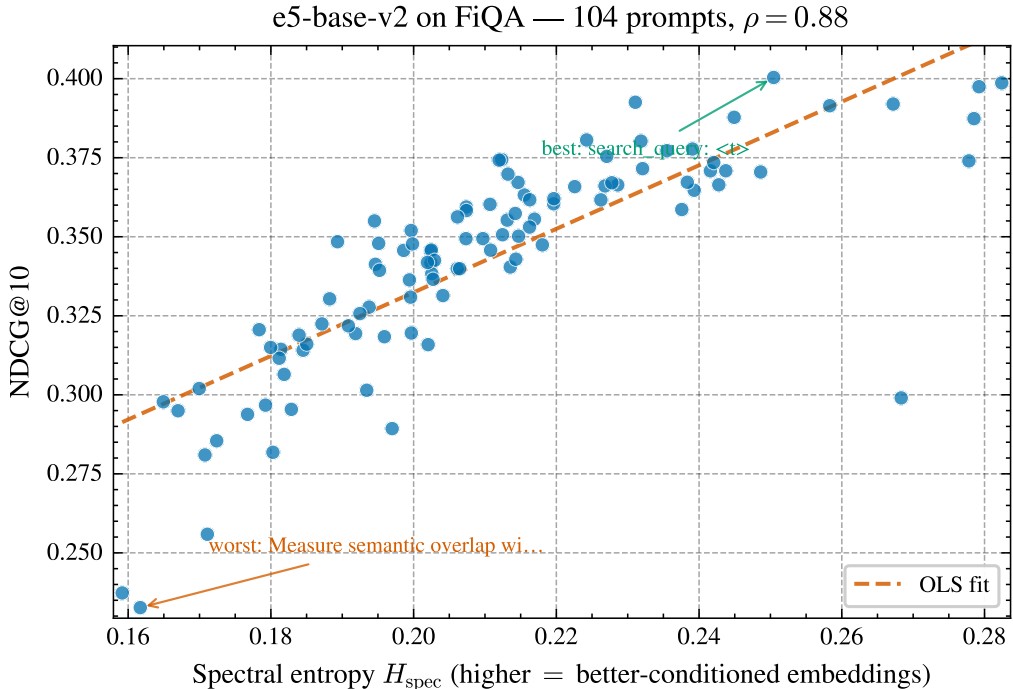

Figure 1: A motivating example on FiQA with E5-base-v2. Each point is one candidate instruction. Spectral entropy computed from 512 proxy texts closely tracks downstream retrieval quality, suggesting that instruction pools can be screened before full retrieval evaluation. The best prefix (`search_query:`) achieves NDCG@10 $= 0.40$ at $H_{spec} = 0.27$, while the worst prefix (`Measure semantic overlap with:`) achieves NDCG@10 $= 0.23$ at $H_{spec} = 0.16$.

Our starting point is geometric rather than lexical. A strong instruction should not only sound retrieval-like in natural language; it should also induce a proxy embedding geometry that is broader, less collapsed, and more discriminative on query-like texts. We therefore hypothesize that the quality of a candidate instruction is reflected in the geometry it induces on a small unlabeled proxy set.

Based on this view, we propose *Instruction Performance Prediction* (IPP), a retrieval-free and label-free procedure for pre-retrieval instruction screening. Given an embedding model, a pool of candidate instructions, and a small *proxy set* (a small unlabeled set of question-like texts used only to probe representation geometry), IPP encodes the proxy texts under each instruction, forms the uncentered second-moment matrix, computes its normalized spectral entropy, and ranks instructions by that score. The method requires no corpus encoding, no query-document scoring, and no relevance labels.

Across 16 embedding models, 17 retrieval datasets, and all 272 model–dataset pairs (full coverage), spectral entropy reaches median oriented Spearman $\rho = 0.806$ against ground-truth instruction rankings. The corresponding median Kendall's $\tau$ is 0.629, median pairwise accuracy is 0.815, exact top-1 hit rate is 21.3%, and median regret@1 is only 0.004 NDCG@10 points. The same evidence also reveals a boundary condition rather than a universal guarantee: when candidate instructions induce very little downstream variance, the ranking problem itself becomes weakly separable, and purely geometric screening should be used with explicit caution.

Our contributions are threefold:

- We formulate *pre-retrieval instruction selection* as a distinct instruction-level prediction problem, separate from post-retrieval QPP.

- We show that the uncentered spectral entropy of proxy embeddings is a simple retrieval-free signal for this problem, and we motivate it geometrically through its connection to pairwise cosine concentration.

- We provide broad empirical evidence together with explicit operating boundaries, matched comparisons to dense-QPP, proxy-source robustness analyses, and evidence on weakly separable instruction pools.

Throughout, we position IPP conservatively: it is not a replacement for final retrieval evaluation, but a first-pass screening tool for the stage before such evaluation is practical.

## 2 Related Work and Positioning

### 2.1 Instruction-Tuned Embeddings and Instruction Selection

Instruction-tuned embedders consolidate diverse text-matching behaviors into a single checkpoint by conditioning the encoder on short natural-language instructions (Su et al., 2023; Wang et al., 2024a;b; Xiao et al., 2023; Chen et al., 2024; Zhang et al., 2025; Hu et al., 2025). They extend the standard bi-encoder retrieval setup (Reimers & Gurevych, 2019; Karpukhin et al., 2020) and are widely evaluated in broad representation benchmarks such as MTEB (Muennighoff et al., 2023). For deployment, this means that a frozen checkpoint still exposes a meaningful control variable: the query-side instruction. The practical question is no longer only how to train an embedder, but also how to choose among the instructions that the embedder already understands.

Recent work on instruction-following retrieval, including FollowIR-style evaluations, studies whether models obey richer retrieval instructions once retrieval is already operational (Weller et al., 2025; Oh et al., 2024; Zhou et al., 2025). That line of work is complementary to ours. It studies instruction following *after* retrieval exists; we study instruction selection *before* instruction-by-instruction retrieval evaluation is practical.

### 2.2 Geometry of Embedding Spaces

The geometry of embedding spaces has long been linked to downstream quality. Prior work studies anisotropy in contextual embeddings, dimensional collapse in contrastive learning, and layer-wise changes in effective rank to explain when embeddings are informative and when they degenerate (Ethayarajh, 2019; Li et al., 2020; Gao et al., 2021; Jing et al., 2022; Rudman & Eickhoff, 2024). Most of that literature is post hoc: geometry is inspected after a model has been trained or after task performance has already been measured. Our use of geometry is different. We treat it as an *a priori* screening signal for a pre-deployment decision.

Among scalar geometric summaries, anisotropy is the closest baseline because it captures directional concentration through mean pairwise cosine similarity. Our question is whether a fuller summary of the uncentered second moment—rather than a first-order concentration statistic alone—is better suited to instruction-level screening. This framing turns embedding geometry from retrospective explanation into pre-retrieval decision support.

### 2.3 Query Performance Prediction and Adjacent Selection Problems

Query Performance Prediction (QPP) estimates query difficulty without access to relevance judgments (Carmel & Yom-Tov, 2010). Recent dense variants replace lexical heuristics with representation perturbations, neighborhood stability, or retrieval-behavior summaries (Faggioli et al., 2023b;a; Arabzadeh et al., 2023; Datta et al., 2023; 2025). PDQPP is the closest published baseline to our setting because it also derives a signal from dense representations (Datta et al., 2025). The operational difference, however, is fundamental: dense-QPP scores *queries* once retrieval exists, whereas IPP ranks *instructions* before retrieval evaluation is available.

Adjacent work on unsupervised retriever selection also targets an earlier choice, but it selects among retrievers rather than instructions and still requires corpus access and heavier inference than our geometry-only screening (Khramtsova et al., 2024). The distinction is not just the score formula; it is the stage of the pipeline at which the method becomes usable.

Table 1: Operational positioning of IPP relative to dense-QPP baselines. The distinction is not only the score formula but also what infrastructure must already exist before the method can be used.

| Family | Unit | Prerequisites | | | Operational use |
|---|---|---|---|---|---|
| | | Corpus | Retrieval | Labels | |
| Classic dense QPP[a] | Query | ✓ | ✓ | ✗ | Estimate per-query difficulty after a retriever exists |
| PDQPP[b] | Query | ✓ | ✓ | ✗ | Score queries from dense retrieval behavior; aggregable for instruction ranking |
| **IPP (ours)** | **Instruction** | ✗ | ✗ | ✗ | **Rank candidate instructions or build a shortlist** *before* **retrieval evaluation** |

[a]Faggioli et al. (2023b;a); Arabzadeh et al. (2023); Datta et al. (2023)   [b]Datta et al. (2025)

Table 1 summarizes the positioning we pursue throughout the paper. IPP is an instruction-level screening primitive for the pre-retrieval stage. It is not intended to replace direct retrieval evaluation once judged queries, corpus access, and stable query budgets become available.

## 3 Problem Formulation, Geometric Motivation, and Screening Method

### 3.1 Formalizing Pre-Retrieval Instruction Selection

Let $\mathcal{M}$ denote a fixed instruction-tuned embedding model and let $\mathcal{P} = \{P_1, \ldots, P_T\}$ be a pool of candidate query-side instructions. For a target retrieval benchmark, the document-side policy is held fixed and only the query-side instruction varies. Let $\mathcal{D}_{\text{proxy}} = \{x_i\}_{i=1}^N$ be a small unlabeled set of query-like proxy texts used only to probe representation geometry. For each instruction $P \in \mathcal{P}$, let $y(P)$ denote the downstream retrieval quality of the model on the target benchmark when queries are encoded with $P$; in our experiments, $y(P)$ is measured primarily by NDCG@10. The goal of IPP is not to predict the absolute value of $y(P)$. Its goal is to recover the *ordering* over instructions induced by $y(P)$ while observing only proxy embeddings.

> **Instruction Performance Prediction (IPP).** Given $(\mathcal{M}, \mathcal{P}, \mathcal{D}_{\text{proxy}})$, produce a ranking $\hat{\pi}$ over $\mathcal{P}$ that is well aligned with the ground-truth ranking $\pi^\star$ induced by $y(P)$, without access to the target corpus, relevance labels, or instruction-by-instruction retrieval evaluation.

We evaluate alignment primarily with *oriented Spearman* $\rho$, where all predictor scores are oriented so that larger values should imply better retrieval. Because deployment decisions are usually top-of-pool decisions rather than full-list correlation judgments, we also report three complementary metrics. Let $\hat{P}_1$ be the top-ranked instruction under $\hat{\pi}$ and let $P^\star = \arg\max_{P \in \mathcal{P}} y(P)$. The exact-hit metric is $\mathbb{I}[\hat{P}_1 = P^\star]$. The absolute regret of the selected instruction is

$$\text{Regret@1} = y(P^\star) - y(\hat{P}_1), \tag{1}$$

reported in target-metric points (for example, NDCG@10 points in the main benchmark). The pairwise accuracy of a predicted ranking is

$$\text{PairAcc}(\hat{\pi}, \pi^\star) = \frac{2}{T(T-1)} \sum_{i<j} \mathbb{I}[(\hat{\pi}(P_i) - \hat{\pi}(P_j))(\pi^\star(P_i) - \pi^\star(P_j)) > 0]. \tag{2}$$

Together, these metrics separate global ranking quality from the practical consequence of picking the top candidate.

### 3.2 Why Uncentered Geometry?

For a fixed instruction $P$, let $\mathbf{Z}^{(P)} \in \mathbb{R}^{N \times d}$ be the matrix of $\ell_2$-normalized proxy embeddings, where $N$ is the number of proxy texts, $d$ is the embedding dimension, and the $i$th row is $z_i^{(P)}$. We study the uncentered

second moment

$$\mathbf{S}^{(P)} = \frac{1}{N}\mathbf{Z}^{(P)\top}\mathbf{Z}^{(P)}. \tag{3}$$

A natural question is whether to work with $\mathbf{S}^{(P)}$ directly or with its centered counterpart $\mathbf{S}_{\mathrm{c}}^{(P)} = N^{-1}\sum_i(z_i^{(P)} - \bar{z}^{(P)})(z_i^{(P)} - \bar{z}^{(P)})^\top$, where $\bar{z}^{(P)} = N^{-1}\sum_i z_i^{(P)}$. We argue that the uncentered form is the natural object here: an instruction can change not only the spread of the proxy embeddings but also their global mean direction, and centering would remove that mean-direction signal by construction. Our claim is therefore not that centered covariance is always inferior in the abstract, but that preserving the instruction-induced global shift is appropriate for the specific problem of instruction screening.

Because the rows of $\mathbf{Z}^{(P)}$ are $\ell_2$-normalized, $\mathrm{tr}(\mathbf{S}^{(P)}) = 1$. The eigenvalues of $\mathbf{S}^{(P)}$ therefore already form a probability distribution over principal directions: they describe how one unit of total embedding energy is allocated across the subspace occupied by the proxy set. A concentrated spectrum means that the embeddings occupy a narrow set of directions, whereas a flatter spectrum indicates broader geometric utilization. The next identity makes the connection to pairwise cosine concentration explicit.

**Proposition 1** (Trace identity for pairwise cosine concentration). *For $\ell_2$-normalized proxy embeddings, the uncentered second moment in Equation 3 satisfies*

$$\mathrm{tr}\Big((\mathbf{S}^{(P)})^2\Big) = \frac{1}{N^2}\sum_{i=1}^N\sum_{j=1}^N\langle z_i^{(P)}, z_j^{(P)}\rangle^2. \tag{4}$$

*Hence, a more concentrated spectrum implies larger mean squared pairwise cosine, while a flatter spectrum implies less cosine concentration across the proxy set.*

*Proof.* Using $\mathbf{S}^{(P)} = N^{-1}\mathbf{Z}^{(P)\top}\mathbf{Z}^{(P)}$,

$$\mathrm{tr}\Big((\mathbf{S}^{(P)})^2\Big) = \frac{1}{N^2}\mathrm{tr}\Big(\mathbf{Z}^{(P)\top}\mathbf{Z}^{(P)}\mathbf{Z}^{(P)\top}\mathbf{Z}^{(P)}\Big) = \frac{1}{N^2}\left\|\mathbf{Z}^{(P)}\mathbf{Z}^{(P)\top}\right\|_F^2.$$

The $(i, j)$ entry of $\mathbf{Z}^{(P)}\mathbf{Z}^{(P)\top}$ is $\langle z_i^{(P)}, z_j^{(P)}\rangle$, which yields Equation 4. $\square$

Proposition 1 does *not* by itself imply retrieval quality. It provides a geometric motivation: if stronger instructions induce a less collapsed proxy geometry, then a scalar that increases with spectral flatness becomes a plausible screening statistic. This interpretation is deliberately bounded. We do not claim that broader geometry is universally better for every embedding task; we claim that, for short query-side instructions in dense retrieval, it is a useful prior for ranking candidate instructions before retrieval evaluation.

### 3.3 From Geometric Insight to Prediction Rule

We operationalize the above intuition with normalized spectral entropy. Let $\lambda_1, \ldots, \lambda_r$ be the nonzero eigenvalues of $\mathbf{S}^{(P)}$, where $r = \min(N, d)$. Since $\mathrm{tr}(\mathbf{S}^{(P)}) = 1$, these eigenvalues already sum to one; equivalently, they define a discrete distribution over principal directions. We therefore define

$$H_{\mathrm{spec}}(P) = \frac{-\sum_{i=1}^r \lambda_i \log \lambda_i}{\log r}. \tag{5}$$

This quantity is a normalized entropy over the spectrum of $\mathbf{S}^{(P)}$, closely related to the effective rank of the proxy embedding matrix (Roy & Vetterli, 2007). Higher $H_{\mathrm{spec}}(P)$ means that the same unit mass is spread more evenly across principal directions.

Our prediction rule is simple: rank instructions by descending $H_{\mathrm{spec}}(P)$. Section 3.4 states the resulting procedure and its inputs; the remaining question is empirical, namely whether this simple geometric proxy is sufficient in practice, and where it stops being reliable.

### 3.4 Workflow and Deployment Role

Figure 2 and Algorithm 1 together summarize the resulting procedure. The method takes three required inputs: a frozen embedder, a candidate instruction pool, and a small query-like proxy set; it produces a full ranking of the pool together with an optional top-$K$ shortlist.

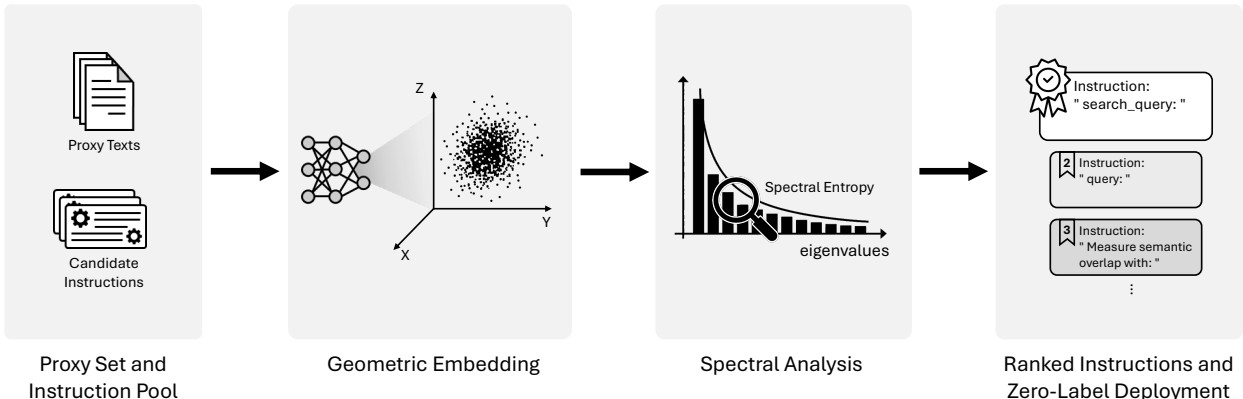

Figure 2: Overview of IPP. Candidate query-side instructions are paired with a small proxy text set and encoded by the embedding model. Spectral analysis of the uncentered second-moment matrix yields a retrieval-free, label-free score for ranking instructions before full retrieval evaluation.

---

**Algorithm 1** Spectral-entropy instruction screening (IPP)

---

**Require:** Embedder $\mathcal{M}$, candidate instructions $\mathcal{P} = \{P_t\}_{t=1}^T$, proxy texts $\mathcal{D}_{\text{proxy}} = \{x_i\}_{i=1}^N$, optional shortlist size $K$

1: **for** each instruction $P_t \in \mathcal{P}$ **do**
2:     Encode the proxy texts and form $\mathbf{Z}^{(P_t)} \in \mathbb{R}^{N \times d}$
3:     $\mathbf{S}^{(P_t)} \leftarrow \frac{1}{N}\mathbf{Z}^{(P_t)\top}\mathbf{Z}^{(P_t)}$
4:     Compute the nonzero eigenvalues of $\mathbf{S}^{(P_t)}$ and score $P_t$ with $H_{\text{spec}}(P_t)$ from Equation 5
5: **end for**
6: Sort $\mathcal{P}$ by decreasing $H_{\text{spec}}$
7: **return** ranked list $\hat{\pi}$ and optional shortlist $\hat{\pi}_{1:K}$

---

The per-instruction cost is dominated by the spectral analysis of $\mathbf{S}^{(P)}$, computable from the nonzero singular values of either $\mathbf{Z}^{(P)}$ or the $N \times N$ Gram matrix $\mathbf{Z}^{(P)}\mathbf{Z}^{(P)\top}$, whichever is smaller; the total work scales linearly in $T$ and involves no corpus encoding, query-document scoring, or relevance supervision. We therefore position IPP as a first-pass screening stage: useful for ranking large instruction pools, choosing a strong default, or producing shortlists for more expensive evaluation, but not a replacement for direct retrieval evaluation once judged queries and stable query budgets become available.

## 4 Empirical Results

We evaluate IPP on a benchmark of 16 embedding models and 17 retrieval datasets, yielding 272 model–dataset pairs (the full $16 \times 17$ grid; an earlier version of this benchmark held out 34 cells as "unsupported" but we fill them in this submission and report results on complete coverage). The model suite spans 13 encoder checkpoints together with 3 decoder-style embedders from the Qwen3 Embedding, GTE-Qwen2-family, and KaLM-Embedding lines (Zhang et al., 2025; Li et al., 2023; Xiao et al., 2023; Hu et al., 2025; Lee et al., 2025; BehnamGhader et al., 2024). The dataset suite combines 6 BEIR benchmarks and 11 MR-TyDi datasets, exposing the method to English retrieval, multilingual retrieval, domain variation, and lower-resource settings (Thakur et al., 2021; Zhang et al., 2021). The candidate pool contains 104 short query-side instructions, and, unless otherwise stated, the proxy set is a fixed collection of $N = 512$ unlabeled question-like texts. For each

model–dataset pair, documents are encoded once under the model's prescribed document-side policy, and only the query-side instruction varies.

Ground-truth instruction quality $y(P)$ is defined primarily by NDCG@10, with NDCG@5 and MRR@10 used for robustness checks. We report oriented Spearman $\rho$, Kendall's $\tau$, exact top-1 hit, absolute regret@1 from Equation 1, and pairwise accuracy from Equation 2. Unless noted otherwise, retrieval-free comparisons use the full 272-pair benchmark; PDQPP is evaluated only on the matched 31-pair subset where its required retrieval-time inputs are available (Datta et al., 2025); the budget study uses a fixed 8-pair illustrative subset; and proxy-source robustness is summarized on 6 representative pairs. Throughout, a *failure* means oriented Spearman $\rho < 0.5$; we use that threshold only as a reporting convention, not as a universal operational cutoff. We organize the evidence around four questions: whether representation geometry can rank instructions without retrieval, whether the signal generalizes across architectures, datasets, and metrics, how robust it is to proxy choice, and how useful it remains under realistic deployment constraints.

## 4.1 Can Representation Geometry Rank Instructions without Retrieval?

We begin with the main retrieval-free comparison against our two closest baselines, oriented anisotropy and TopoScore (defined in Appendix B).[1] Table 2 gives the short answer. On the full 272-pair benchmark, ranking the 104 candidate instructions by spectral entropy yields median oriented Spearman $\rho = 0.806$, median Kendall's $\tau = 0.629$, median pairwise accuracy 0.815, and a failure rate of 14.3% under the paper-wide definition. The exact top-1 hit rate is 21.3%, and the median regret@1 is 0.004 NDCG@10 points against the ground-truth ranking.

The 21.3% top-1 hit rate is intentionally strict for a 104-instruction pool, as random selection would succeed only $1/104 = 0.96\%$ of the time, but the more deployment-relevant number is regret@1: the selected instruction is typically within half a percentage point of NDCG@10 of the oracle choice.

Table 2: Headline retrieval-free predictor comparison on the 272-pair benchmark. Failure: oriented $\rho < 0.5$. Regret@1 in NDCG@10 points. **Bold** marks the best value per column.

| Predictor | Signal type | Median $\rho$ | Median $\tau$ | Fail rate | Top-1 hit | Regret@1 |
|---|---|---|---|---|---|---|
| **Spectral entropy (ours)** | spectrum | **0.806** | **0.629** | **14.3%** | 21.3% | **0.004** |
| Oriented anisotropy | 1st-order | 0.800 | 0.617 | 14.7% | **23.5%** | **0.004** |
| TopoScore | neighborhood | 0.536 | 0.376 | 47.8% | 8.5% | 0.021 |

The separation from TopoScore is large, while the comparison to oriented anisotropy is genuinely close. Paired Wilcoxon tests show a clear gap to TopoScore (median $\Delta\rho = 0.051$, $p = 1.87 \times 10^{-29}$, effect size $r = 0.68$), but a much smaller gap to oriented anisotropy (median $\Delta\rho = 0.002$, $p = 3.17 \times 10^{-10}$, $r = 0.41$). Read fairly, the two geometric predictors split the metrics: oriented anisotropy is marginally better on top-of-pool decisions (23.5% vs. 21.3% top-1 hit; regret@1 tied at 0.004), whereas spectral entropy is marginally better on global ranking ($\rho$, $\tau$, pairwise accuracy) with a slightly lower failure rate (14.3% vs. 14.7%).

We nevertheless adopt spectral entropy as the main score for two reasons, one empirical and one principled. Empirically, our formal target is instruction-ranking recovery, and on that target spectral entropy is stronger. Principled, Section 3.2 motivates it through the full uncentered spectrum, whereas anisotropy summarizes only mean pairwise cosine concentration — a strictly first-order statistic that Proposition 1 shows is one moment of the same underlying object. The takeaway is not that spectral entropy dominates every geometric scalar, but that a simple retrieval-free spectral statistic already carries enough signal to rank instructions reliably across a broad benchmark. Appendix G reports a broader predictor landscape together with a few-shot calibration curve. Those results suggest that labels add only modest headroom once geometry features are already available, which is why we treat supervised composites as contextual upper-bound evidence rather than as the main method.

---

[1] Oriented anisotropy scores an instruction by the negative mean pairwise cosine concentration of its proxy embeddings; TopoScore scores it by how well its proxy-space neighborhood structure agrees with a reference structure induced by a fixed set of anchor instructions.

## 4.2 Does the Signal Generalize Across Architectures, Domains, and Metrics?

The next question is whether the signal survives beyond a few especially easy model–dataset pairs. Figure 3 shows that it does. Most cells are strongly positive, and the weaker cells are structured rather than random: they cluster in Bengali, Telugu, Thai, and a small number of long-document BEIR subsets. These are also the settings in which instruction-induced retrieval variance is small and the ranking problem itself is less separable.

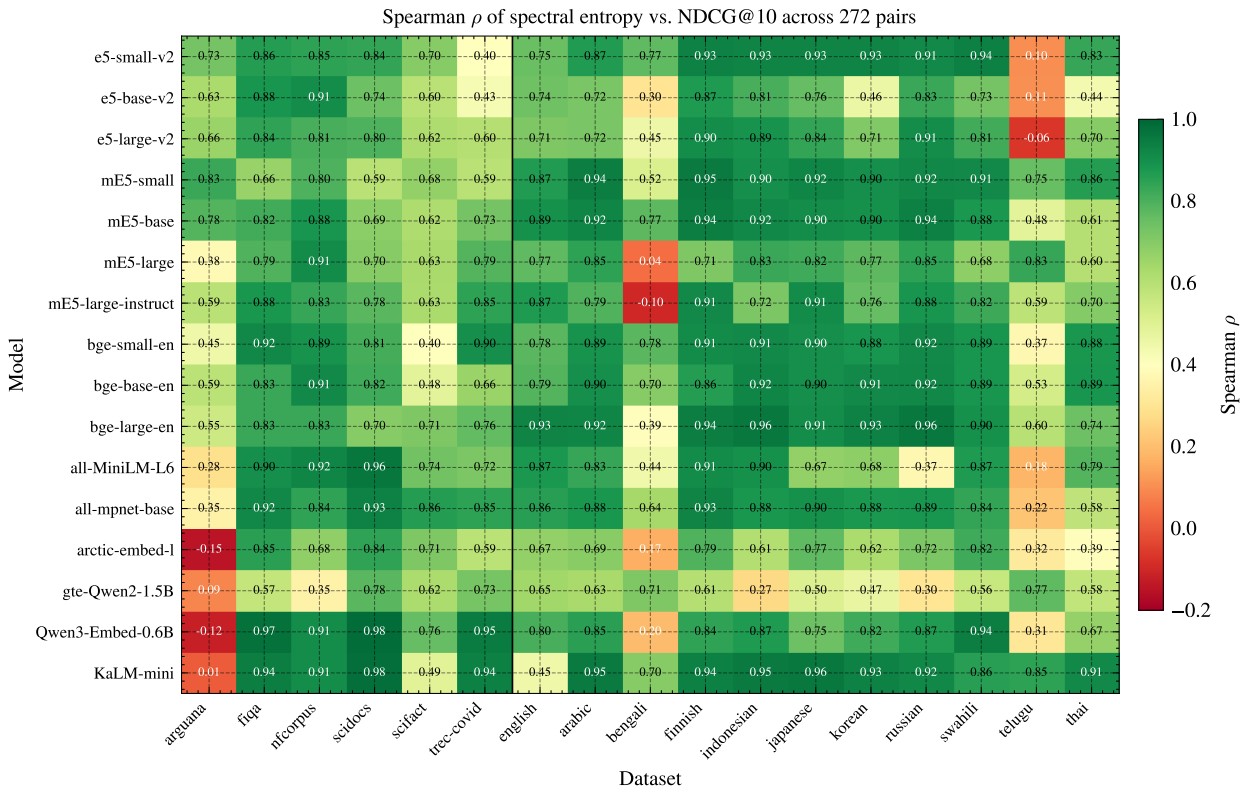

Figure 3: Spectral-entropy ranking quality across the 272-pair benchmark (full coverage). Each cell reports Spearman $\rho$ between predicted and ground-truth instruction rankings (target: NDCG@10). Gray cells mark the 34 model–language combinations excluded as unsupported. Most cells are strong; weaker cells concentrate in lower-resource languages and a small number of instruction-insensitive datasets.

Table 3 summarizes the same pattern at the family level. The effect is not confined to one architecture class: the encoder family reaches median $\rho$ = 0.813, while the three decoder / instruction-tuned embedders (`gte-Qwen2-1.5B-instruct`, `Qwen3-Embedding-0.6B`, `KaLM-embedding-multilingual-mini-instruct-v2.5`) still reach 0.766. MR-TyDi is the strongest dataset family overall. The weakest stratum is the four lower-resource languages in MR-TyDi: Bengali, Swahili, Telugu, and Thai. For these languages, both the median correlation and the elevated failure rate suggest that the instruction-ranking problem itself is less separable there, not that the predictor is failing randomly. The effect is broad, but not uniform.

Table 3: Stratified ranking quality of spectral entropy on the 272-pair benchmark. Failure: oriented $\rho < 0.5$.

| Group | $n$ pairs | Median $\rho$ | Fail rate |
|---|---|---|---|
| Encoder family | 221 | 0.813 | 12.7% |
| Decoder / instruct pairs | 51 | 0.766 | 21.6% |
| BEIR English datasets | 96 | 0.772 | 14.6% |
| MR-TyDi datasets | 176 | 0.831 | 14.2% |
| Lower-resource languages | 64 | 0.696 | 29.7% |

The aggregate result is also not driven by one especially easy family. Appendix F reports a leave-one-family-out audit on the 272-pair benchmark, showing that removing any single model family or dataset family shifts the median correlation but does not collapse it. The result is also not tied to one downstream metric: the median $\rho$ against the ground-truth ranking remains 0.799 for NDCG@5 and 0.778 for MRR@10, within 0.03 of the 0.806 obtained against NDCG@10. This is consistent with the interpretation that spectral entropy screens instruction-induced embedding quality rather than overfitting a single evaluation measure.

### 4.3   How Robust Is the Predictor to Proxy Choice?

A natural concern is that the method might depend on a carefully tuned in-domain proxy source. The data support a narrower and more useful claim. The proxy set does not need to be in-domain, but it does need to be *query-like*. Table 4 shows this on six representative pairs: generic questions and out-of-domain queries remain close to in-domain queries, whereas declarative Wikipedia sentences are substantially weaker. This makes the cold-start use case plausible without claiming that proxy choice is irrelevant.

Table 4: Proxy-source robustness across six representative model–dataset pairs at $N = 512$, target NDCG@10. Question-like proxy texts remain close to in-domain queries, whereas declarative Wikipedia sentences are substantially weaker.

| Proxy source | Mean $\rho$ | Median $\rho$ |
|---|---|---|
| In-domain queries | 0.74 | 0.72 |
| OOD queries | 0.72 | 0.72 |
| Generic questions | 0.71 | 0.71 |
| Wikipedia sentences | 0.50 | 0.54 |

A second robustness result is that the ranking is stable under repeated proxy resampling. Across the preserved 10-seed audit, the per-pair standard deviation of Spearman $\rho$ has median 0.002, so the screening decision is not an artifact of one fortunate proxy draw. Appendix C provides the detailed proxy-source and resampling panels.

### 4.4   How Useful Is IPP under Deployment Constraints?

The practical claim of the paper is not merely that spectral entropy correlates with retrieval quality. It is that spectral entropy is useful precisely when direct retrieval evaluation is unavailable, delayed, or too expensive to run for every candidate instruction. We test that claim in two ways.

**Comparison to the closest dense-QPP baseline.**   PDQPP is the closest prior baseline because it also derives a signal from dense representations (Datta et al., 2025). The comparison is deliberately asymmetric in PDQPP's favor: it sees the full corpus and runs retrieval-time computations, whereas IPP never touches either. Even under this asymmetry, spectral entropy remains competitive on the matched 31-pair BEIR-English subset where both methods ran: median $\rho = 0.737$ (95% CI [0.661, 0.805]) for IPP against 0.718 ([0.642, 0.840]) for PDQPP with mean aggregation, with 10,000-resample bootstrap CIs and a paired difference that is not statistically significant ($p = 0.399$). We do not claim to replace post-retrieval QPP; the point is that a pre-retrieval geometric proxy remains competitive with a post-retrieval dense-QPP baseline while requiring strictly less information.

**Budget–quality trade-off.**   A screening method must also justify itself once small amounts of judged data start to arrive. Figure 4 compares spectral-entropy screening against budget-matched retrieval evaluation on a fixed 8-pair illustrative subset. At very small budgets ($\leq 64$ encodes per candidate instruction), mini-retrieval is noisy and spectral-entropy screening yields lower regret. As the query budget grows beyond roughly 128 judged queries, direct retrieval evaluation appropriately overtakes screening. This is the intended division of labor: use geometry first, then hand off to direct evaluation once judged data are sufficiently stable to support it.

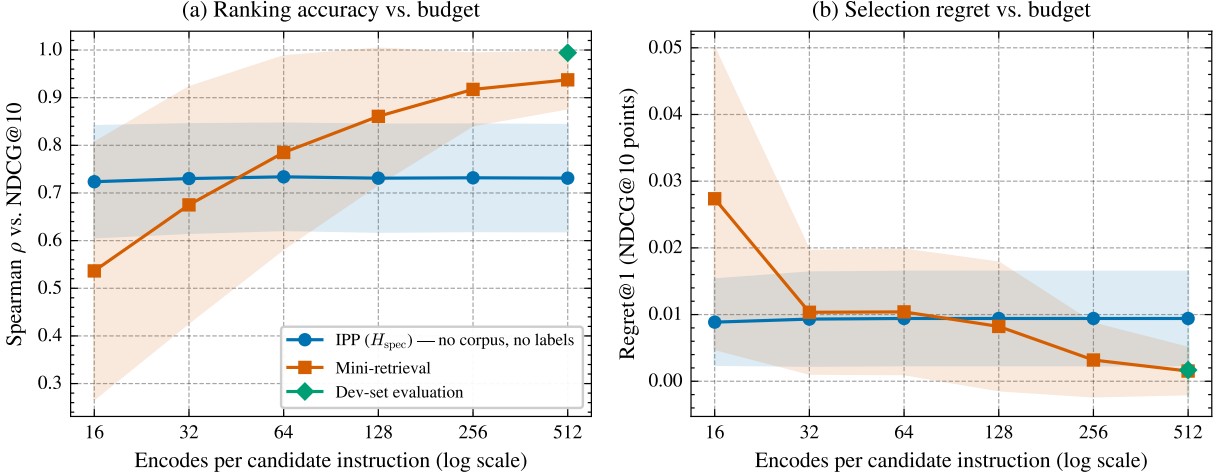

Figure 4: Budget–quality trade-off on a fixed 8-pair illustrative subset. The x-axis reports encodes per candidate instruction (log scale); shaded bands show mean ± standard deviation over three resampling seeds. At small budgets, spectral-entropy screening yields lower regret than mini-retrieval; once enough judged queries are available, direct retrieval evaluation overtakes it.

# 5 What IPP Measures, When It Fails, and What It Does Not Claim

## 5.1 What the Score Is Measuring

Figure 5 helps interpret the score itself. On the 9-pair geometry audit, centering the proxy embeddings before spectral analysis reduces the median ranking correlation from 0.681 to 0.286 ($p = 0.0039$) and increases the median pairwise inversion rate from 0.255 to 0.405. For this task, the mean direction induced by the instruction is therefore not nuisance structure; it is part of the usable signal. The same figure also shows that most of the raw signal is retained in a relatively small number of leading singular directions, which is more consistent with a structured geometric effect than with diffuse high-dimensional noise. Appendix E adds a complementary paraphrase audit showing that the signal remains informative even within clusters of closely related instructions.

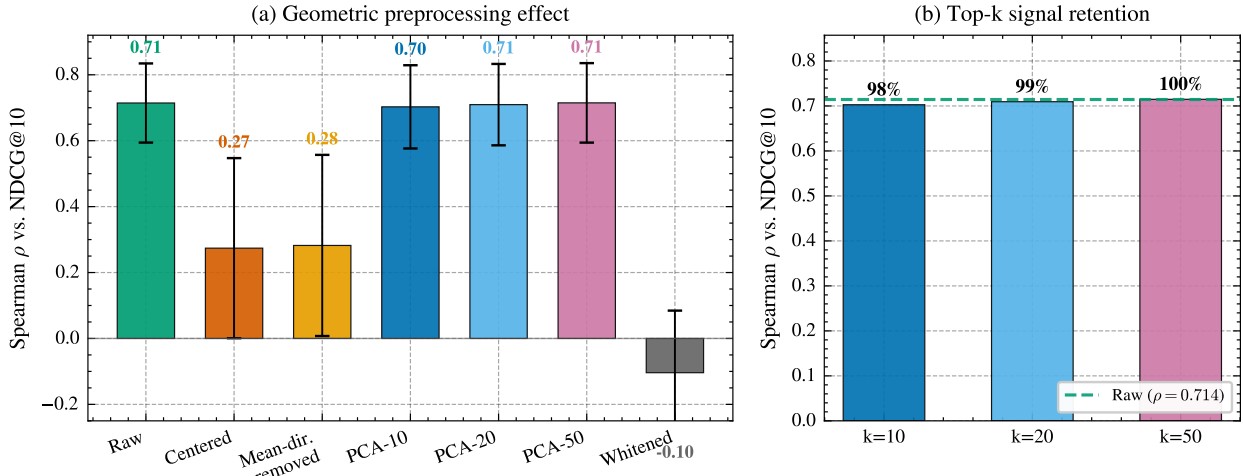

Figure 5: Geometry audit for the design choice in Section 3.2. Panel (a) compares spectral-entropy variants on the 9-pair geometry audit, and panel (b) shows that most of the raw signal is retained in a small number of leading singular directions.

Figure 6 adds a complementary mechanistic view. Across three encoder models and three decoder-based embedders, stronger instructions tend to preserve larger entropy gaps in deeper layers and stronger layer-wise predictive power. We read this as evidence consistent with an instruction-induced geometry explanation rather than a purely superficial metadata effect. The claim should remain modest: these probes support the geometric interpretation, but they do not establish a unique mechanism.

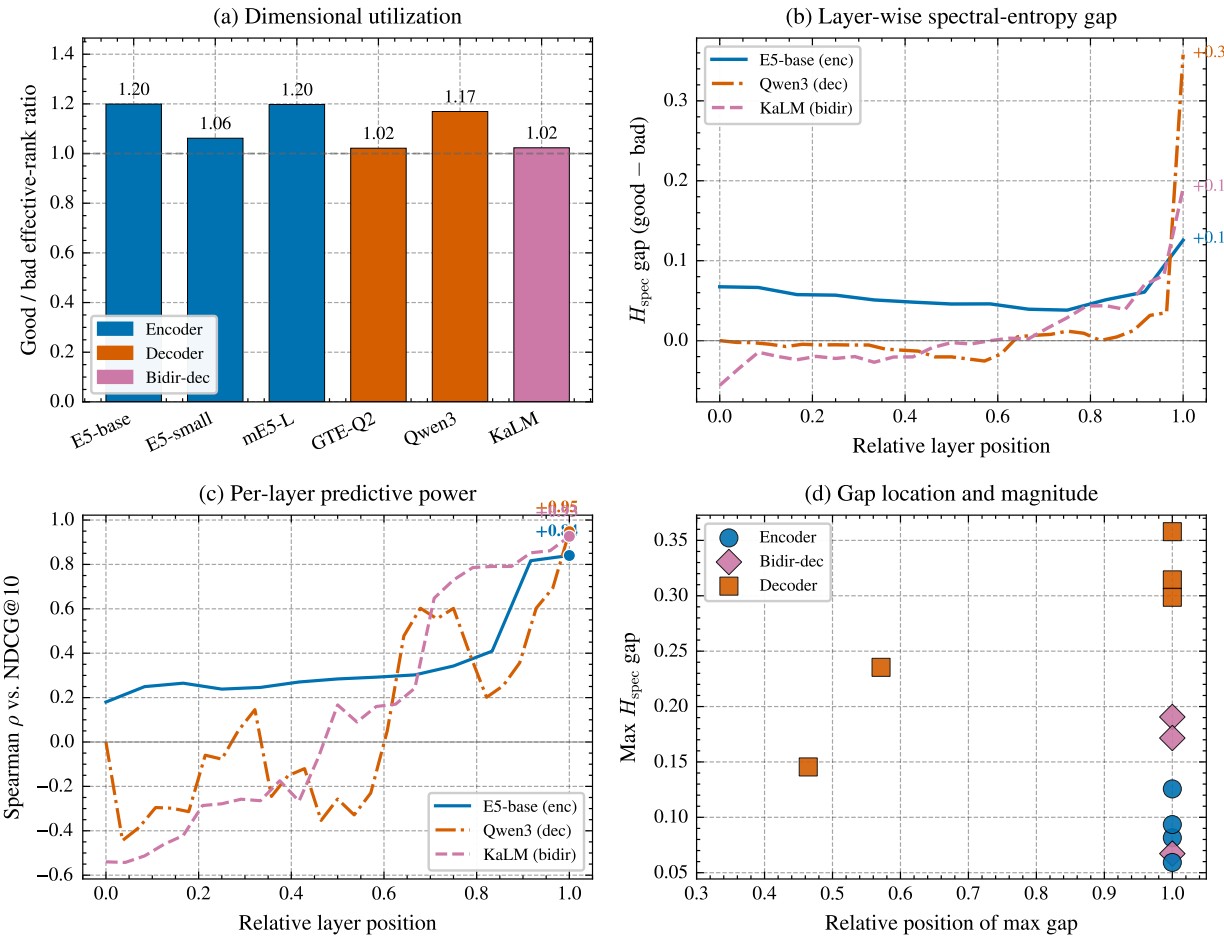

Figure 6: Mechanistic probe suite across six representative embedders. The panels summarize dimensional utilization, layer-wise spectral-entropy gaps, per-layer predictive power, and the location / magnitude of the strongest entropy gap. The panels most directly relevant to the paper's interpretation are the sustained layer-wise entropy gaps and the deeper-layer predictive power for stronger instructions.

Taken together, these analyses suggest a narrow interpretation of the score. IPP is not estimating instruction quality in the abstract. It is estimating whether a short query-side instruction induces a retrieval-ready, relatively uncollapsed proxy geometry on question-like texts. That is the paper's central insight: a pre-retrieval decision can be informed by a measurable property of instruction-induced representation geometry.

## 5.2 When the Method Fails

The most useful negative result is that weak cases are not random failures of one scoring rule. They cluster in the same parts of the benchmark that already look comparatively instruction-insensitive in the main results. Under the paper-wide reporting convention of oriented $\rho < 0.5$, spectral entropy fails on 14.3% of the full 272-pair benchmark (Table 2), but the failures are concentrated rather than diffuse. Table 3 shows fail rates of 29.7% for lower-resource languages and 21.6% for the decoder / instruct pairs, and Figure 3 shows the corresponding pockets of weaker cells in Bengali, Telugu, Thai, and a small number of BEIR subsets. This

concentration suggests a property of the task instances, not random noise: in these settings the candidate instructions often induce only modest downstream differences, so recovering a stable full order is intrinsically harder.

This framing matters for interpretation. In a weakly separable pool, exact top-1 recovery becomes a harsh criterion, whereas low regret is often the more informative operational signal. The practical recommendation is therefore conservative. Use IPP to screen a large pool and identify a shortlist, but hand off to mini-retrieval or direct evaluation when score gaps are small, when the task family is known to be instruction-insensitive, or when the shortlisted candidates remain tightly clustered after screening. Optional guardrails such as instruction sensitivity, anchor consensus, and spectral-entropy dispersion are best read as heuristic warning lights for this hand-off, not as calibrated guarantees.

### 5.3 What the Paper Does Not Claim

We close by stating the paper's limits explicitly. First, IPP is not a general prompt-optimization method. It targets short query-side instructions for dense embedding retrieval. The results should not be read as evidence that the same score will optimize long natural-language system prompts, multi-turn prompting behavior, or instruction following outside retrieval.

Second, the method is not proxy-free. The evidence shows that the proxy texts do not need to be in-domain, but they should remain question-like. This is a manageable assumption for cold-start deployment, not a claim that any arbitrary text source can be substituted without loss.

Third, the decoder-side evidence is encouraging but still limited. The transfer results justify the claim that the phenomenon is not encoder-only within the tested benchmark, but they do not justify universal decoder coverage.

Fourth, any optional guardrails are heuristics rather than calibrated guarantees. They are useful only insofar as they warn that a pool may be low-variance or weakly separable. When those warning lights trigger, the correct response is not to stack more geometry heuristics; it is to fall back to direct retrieval evaluation.

## 6 Conclusion

We studied pre-retrieval instruction selection for instruction-tuned embedding models and argued that it should be treated as a distinct instruction-level prediction problem rather than as a variant of post-retrieval QPP. The main empirical finding is that the uncentered spectral entropy of proxy embeddings provides a strong zero-label signal for ranking candidate instructions before corpus-side evaluation is practical. Across the full 272-pair benchmark, that signal is strong both as a ranking measure and as a top-of-pool decision rule. Under these conditions, early instruction screening can be grounded in a measurable geometric proxy rather than handled entirely by exhaustive trial and error.

The paper also makes a bounded claim. IPP is useful when direct retrieval evaluation is unavailable, delayed, or too expensive to run for every candidate instruction. It is not a replacement for final retrieval evaluation, and its reliability weakens when candidate instructions induce very little downstream variance. The operating boundary is therefore part of the contribution, not an afterthought.

Practically, IPP is best viewed as a first-pass selector. It can rapidly screen large instruction pools, identify strong defaults, and decide which candidates deserve more expensive testing. Once judged data and stable query budgets become available, direct retrieval evaluation should take over.

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

## A  Additional Protocol Details

Figure 4 uses three budget-matched selectors on the fixed 8-pair scenario study. IPP scores each candidate instruction from $N = 512$ proxy-text encodes and no retrieval stack. *Mini-Retrieval* subsamples $k$ judged queries together with a 1000-document corpus slice and computes NDCG@10 on that reduced retrieval problem, whereas *Dev-Set Eval* uses the same judged-query budgets against the full corpus. The x-axis therefore reports query or proxy encodes per candidate instruction; one-time corpus indexing is excluded from the budget axis even though corpus access remains a prerequisite for the retrieval baselines.

## B  Formal Definitions of Auxiliary Signals

This appendix records the auxiliary quantities referenced in the main text. All definitions operate on the proxy embedding matrix $\mathbf{Z}^{(P)}$ for a candidate instruction $P$ and on a fixed set of anchor instructions $\{A_m\}_{m=1}^{L}$ that are disjoint from the candidate instruction pool.

**Topology alignment / TopoScore.** For each anchor instruction and candidate instruction, we build soft neighborhood distributions over the proxy texts using multi-scale $k$-NN graphs with $k \in \{5, 10, 20, 50\}$. Let $\hat{p}_{ij}$ denote the consensus neighborhood distribution induced by the anchor instructions and let $p_{ij}^{(P)}$ denote the corresponding distribution under candidate instruction $P$. We define

$$\text{TopoScore}(P) = -\frac{1}{N} \sum_{i=1}^{N} \sum_{j=1}^{N} \hat{p}_{ij} \log p_{ij}^{(P)}, \tag{6}$$

so that larger values indicate better agreement with the anchor consensus.

**Instruction sensitivity.** Let $J(\cdot, \cdot)$ denote Jaccard overlap between two $k$-NN sets. Instruction sensitivity is

$$\Delta(P) = 1 - \frac{1}{L} \sum_{m=1}^{L} J(k\text{-NN}(P), k\text{-NN}(A_m)). \tag{7}$$

It measures how strongly a candidate instruction perturbs local neighborhood structure relative to the anchors.

**Instruction effect.** Let $\mathbf{z}_i^{(\emptyset)}$ be the embedding of proxy text $x_i$ under the no-prefix baseline. Instruction effect is

$$E(P) = 1 - \frac{1}{N} \sum_{i=1}^{N} \langle z_i^{(P)}, z_i^{(\emptyset)} \rangle. \tag{8}$$

This captures the magnitude of the instruction-induced shift away from the no-prefix embedding geometry.

**Anchor consensus.** Anchor consensus is the mean pairwise neighborhood agreement among the fixed anchor instructions. We use it only as a reliability signal for the instruction-ranking problem itself, not as a scoring rule for instruction selection.

## C  Additional Proxy and Stability Results

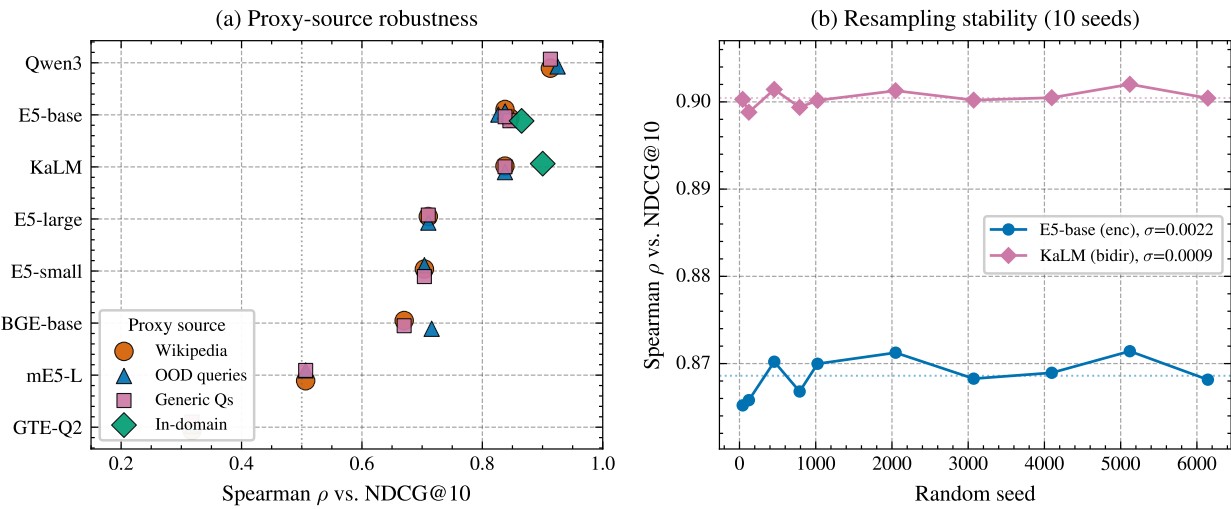

Figure 7: Additional robustness analyses. Panel (a) shows representative model–dataset pairs across proxy sources, and panel (b) shows representative pairs with 10 proxy resamples each. Question-like proxies are robust to domain mismatch, and proxy-set resampling introduces very little variance for both encoder and decoder-based embedders.

Figure 7 expands the robustness story in two ways. First, it shows that generic questions and out-of-domain queries stay close to in-domain query proxies on representative pairs, which is why the main text recommends question-like proxy sets. Second, it shows that the spectral-entropy ranking is stable under proxy-set resampling, indicating that the method is not overly sensitive to one particular random draw of 512 proxy texts.

## D Supplementary Geometry Audit

Table 5: Numerical summary of the centered-vs.-uncentered audit using median summaries throughout. The raw uncentered variant dominates both by rank correlation and by pairwise inversion rate.

| Comparison | $n$ | Raw median | Centered median | $p$ |
|---|---|---|---|---|
| Variant $\rho$ | 9 | 0.681 | 0.286 | 0.0039 |
| Pairwise inversion rate | 9 | 0.255 | 0.405 | 0.0039 |

## E Paraphrase Sensitivity Audit

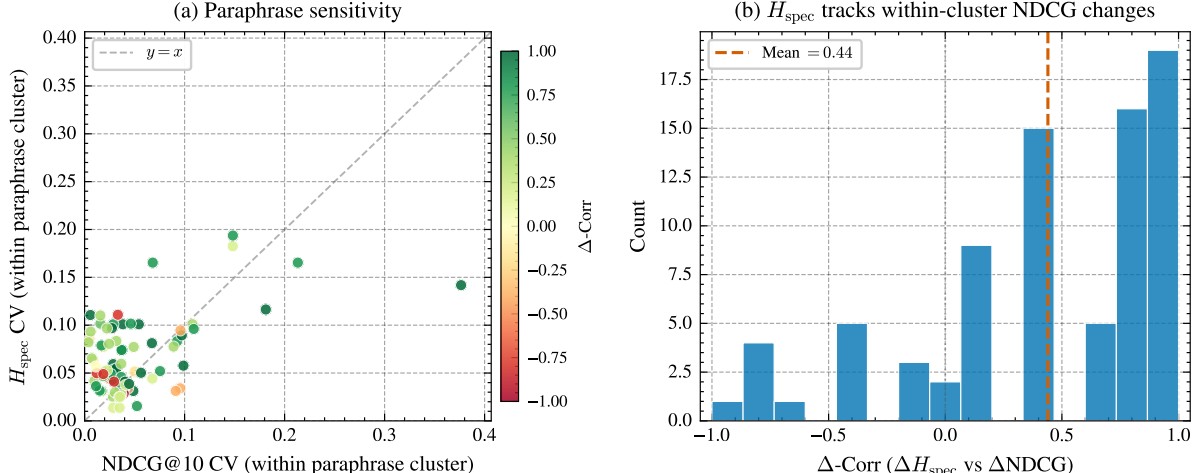

Figure 8: Paraphrase sensitivity audit. Panel (a) compares within-cluster variation of spectral entropy and NDCG@10 across paraphrase groups, and panel (b) summarizes the correlation between within-cluster changes in $H_{\text{spec}}$ and downstream performance. The average association remains positive, suggesting that the signal is not explained only by coarse wording differences between obviously dissimilar instructions.

Figure 8 provides a complementary robustness check. Even within paraphrase clusters, changes in spectral entropy tend to track changes in retrieval quality, with mean $\Delta$-correlation 0.44. This does not eliminate lexical confounding, but it suggests that the score is not driven solely by coarse wording differences between obviously dissimilar instructions.

## F  Leave-One-Family-Out Robustness

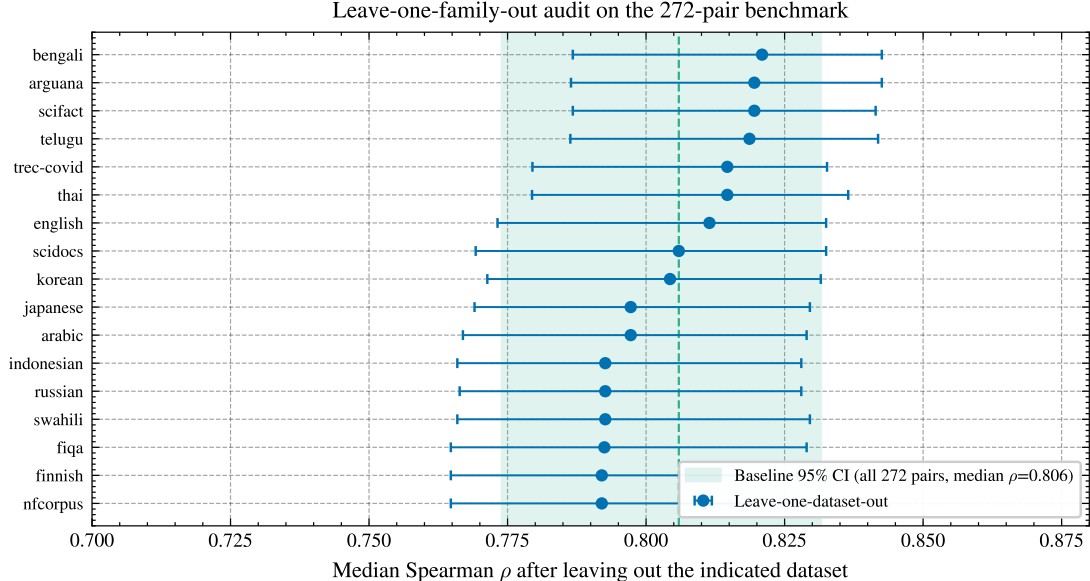

Figure 9: Leave-one-dataset-out audit on the 272-pair benchmark. The shaded band marks the 95% bootstrap interval of the all-pairs median correlation. Removing any single dataset leaves the median correlation within a relatively narrow range, arguing against the main result being driven by one especially easy dataset.

Table 6: Coarse family-removal summary on the 272-pair benchmark (full coverage). These removals complement Figure 9 by showing that the result also remains intact across encoder/decoder and BEIR/MR-TyDi partitions.

| Held-out family | Remaining $n$ | Median $\rho$ | 95% CI |
|---|---|---|---|
| None (baseline) | 272 | 0.806 | [0.774, 0.832] |
| Remove encoder family | 51 | 0.766 | [0.647, 0.859] |
| Remove decoder/instruct | 221 | 0.813 | [0.775, 0.833] |
| Remove BEIR English | 176 | 0.831 | [0.783, 0.863] |
| Remove MR-TyDi | 96 | 0.772 | [0.710, 0.816] |

# G   Composite Context and Additional Predictor Comparisons

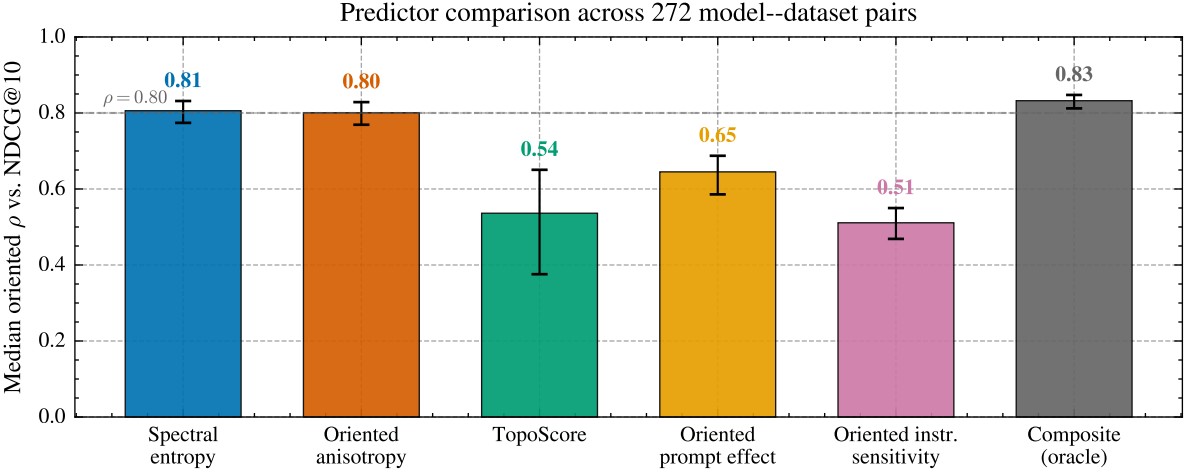

*Composite uses labeled NDCG for weighting (oracle upper bound, not deployable).*

Figure 10: Broader predictor landscape on the 272-pair benchmark (full coverage). The oracle composite is shown only as upper-context and is not deployable without labels. The main unsupervised takeaway is that spectral entropy and oriented anisotropy form the strongest retrieval-free signals, with TopoScore clearly behind.

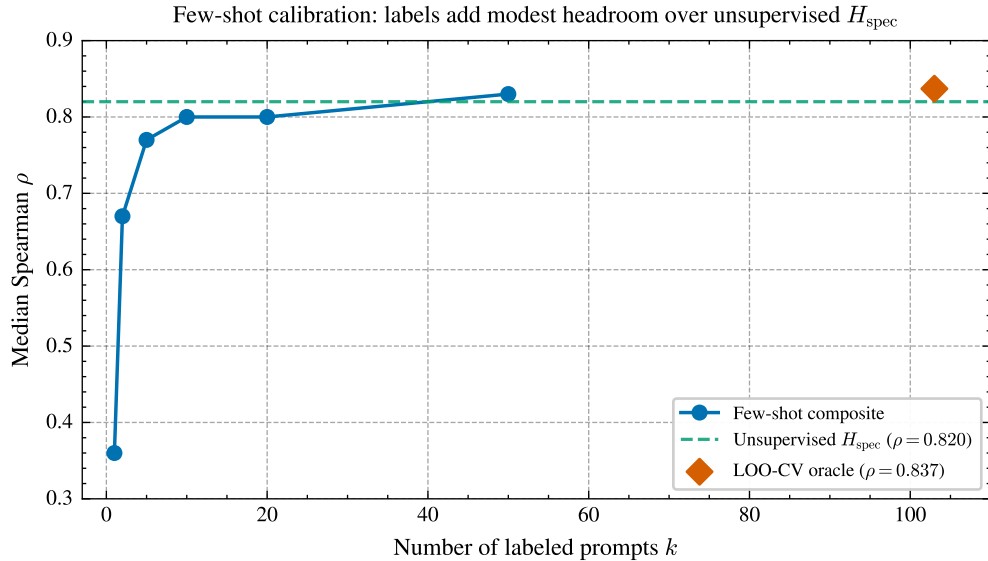

Figure 11: Few-shot calibration context for the labeled composite. Correlation improves only gradually as more instruction outcomes are revealed, leaving the unsupervised spectral-entropy baseline competitive until larger label budgets are available.

Table 7: Paired comparison statistics on the 272-pair benchmark (full coverage). The gap from spectral entropy to TopoScore is large, whereas the gap to oriented anisotropy is small even when statistically detectable.

| Comparison | Median $\Delta\rho$ | $p$ | Effect size $r$ |
| --- | --- | --- | --- |
| SE vs. TopoScore | 0.051 | $1.87 \times 10^{-29}$ | 0.683 |
| SE vs. oriented anisotropy | 0.002 | $1.07 \times 10^{-13}$ | 0.451 |

