# OpenReview forum: "Retrieval-Free Instruction Selection for Instruction-Tuned Embedding Models via Uncentered Spectral Entropy"
_TMLR — Under review for TMLR_

### Review · Reviewer_WgiH · 2026-05-31

**Summary Of Contributions:**

This paper studies pre-retrieval instruction selection for instruction-tuned embedding models. It proposes IPP, a retrieval-free method that ranks candidate query-side instructions using the spectral entropy of proxy query embeddings. The main result is that this simple geometric signal correlates well with downstream retrieval performance across many models and datasets.

Strengths:
1. The paper shows that query-side instruction choice can affect retrieval performance, while full retrieval evaluation may require corpus access, relevance labels, and repeated retrieval runs.
2. The proposed method ranks candidate instructions without using corpus access, relevance labels, or retrieval results.
3. The experiments cover multiple embedding models, retrieval datasets, and evaluation metrics.

Weakness:
The main concern is that some practical details of the proxy setup remain under-characterized. Although the paper includes some analysis of proxy sources, it provides limited guidance on how to construct query-like proxy texts in practice and how sensitive the method is to the proxy budget. This matters because the method is motivated by cold-start and low-resource deployment settings.

**Audience:**

Yes

**Audience Explanation:**

Yes.  The problem is practical, and the proposed method is lightweight enough to be relevant for real deployment pipelines.

**Broader Impact Concerns:**

There are no obvious ethical concerns regarding the topic and the experiments.

**Claims And Evidence:**

Yes

**Claims Explanation:**

Yes. The main empirical claims are supported by a broad set of experiments.

**Requested Changes:**

I hope the authors could clarify the construction of the proxy texts and add a small proxy-size sensitivity analysis. This is important for judging whether the method is actually useful in cold-start or low-budget settings.

---

> ### Author Response · Authors · 2026-06-02
> **Response to Reviewer WgiH**
>
> We thank the reviewer for the positive and constructive assessment of our work. We especially appreciate the recognition that IPP is practical and lightweight for cold-start deployment. We agree that proxy-text construction and sensitivity to the proxy budget deserve clearer treatment, since these details are important for judging the method’s usefulness in low-resource settings.
>
>
>
> **Proxy construction.** We will revise the paper to make the proxy construction protocol explicit. In our experiments, (D_{\text{proxy}}) is a fixed set of unlabeled, query-like texts shared across all candidate instructions. It is used only to probe instruction-induced embedding geometry and does not require access to the target corpus, relevance labels, retrieval outputs, or instruction-by-instruction retrieval evaluation.
>
>
>
> In practice, we recommend the following recipe. If unlabeled target-domain queries are available, they are the natural default. Otherwise, out-of-domain search/query logs or generic question-like texts are suitable alternatives. We will also clarify that arbitrary declarative text is a weaker default proxy source. This recommendation is supported by our existing proxy-source analysis at (N=512): in-domain queries, OOD queries, and generic questions obtain similar mean/median Spearman correlations of 0.74/0.72, 0.72/0.72, and 0.71/0.71, respectively, whereas Wikipedia-style declarative sentences are substantially weaker, with mean/median correlations of 0.50/0.54. We will report these numbers more explicitly in the discussion and add representative examples and selection guidance.
>
>
>
> **Proxy-size sensitivity.** To address the reviewer’s concern about proxy budget, we have conducted a controlled proxy-size sensitivity probe. The goal of this experiment is to isolate the effect of (N), so we fix the model, dataset, candidate instruction pool, and proxy source, and vary only the number of proxy texts. Specifically, we use the representative e5-base-v2 × FiQA setting with the full 104-instruction pool, and recompute (H_{\text{spec}}) for each proxy size
>
> [
>
> N \in {32,64,128,256,512}.
>
> ]
>
> We then evaluate Spearman (\rho) between the IPP ranking and the ground-truth NDCG@10 instruction ranking.
>
>
>
> The results show that IPP is highly stable once a small query-like proxy set is available:
>
>
>
> |                                Proxy size (N) |    32 |    64 |   128 |   256 |   512 |
>
> | --------------------------------------------: | ----: | ----: | ----: | ----: | ----: |
>
> | (H_{\text{spec}}) Spearman (\rho) vs. NDCG@10 | 0.807 | 0.811 | 0.811 | 0.813 | 0.810 |
>
>
>
> Across (N=32) to (512), the total variation in Spearman (\rho) is less than 0.01, and the (N=32) result is already essentially identical to the (N=512) default. This indicates that IPP does not require a large proxy budget in this representative setting; a few dozen query-like proxy texts are already sufficient to recover a stable instruction ranking.
>
>
>
> This sensitivity probe complements the existing (N=512) resampling audit, where the median per-pair standard deviation of Spearman (\rho) across 10 proxy resamples is approximately 0.002. Together, these results support the intended cold-start use case: IPP is not proxy-free, but it only requires a small unlabeled set of query-like texts, and our proxy-source robustness results show that these texts need not be in-domain.
>
>
>
> We will add the proxy construction recipe and the proxy-size sensitivity table to the revised manuscript, together with a short discussion clarifying the operating recommendation: use target-domain queries when available, otherwise use OOD or generic question-like texts, and avoid long declarative document sentences as the default proxy source. These revisions clarify the practical setup without changing the method or the main conclusions.

---

### Review · Reviewer_SK32 · 2026-06-19

**Summary Of Contributions:**

This paper studies a practical problem in instruction-tuned embedding models: given a fixed pretrained embedding model, how should we choose the query-side instruction for retrieval? The key observation is that the instruction is not merely cosmetic. Even with the same query, different prefixes such as “search_query:” or “Find relevant documents for:” can change the query embedding and affect retrieval performance.

The main contribution is a retrieval-free and label-free screening method for instruction selection. Instead of evaluating every candidate instruction on a full corpus with relevance labels, the proposed method embeds a small set of unlabeled query-like proxy texts under each instruction and measures the geometry of the resulting embeddings using uncentered spectral entropy. The intuition is that better instructions tend to produce a less collapsed and more informative embedding space, so they can be ranked before full retrieval evaluation is available.

A key strength is that the method targets a realistic cold-start setting where corpus access, labels, or large-scale evaluation may not yet be available. The empirical evaluation is also broad, covering many embedding models, datasets, and candidate instructions, and the results show strong correlation with downstream retrieval quality and low regret in selecting top instructions.

**Additional Comments:**

NA

**Audience:**

Yes

**Audience Explanation:**

The paper addresses a practical issue in modern retrieval and RAG systems: how to choose the query-side instruction for an instruction-tuned embedding model when full retrieval evaluation is not yet available. This is relevant to researchers working on dense retrieval, embedding models, instruction tuning, retrieval-augmented generation, and deployment-time model selection.

The paper is also interesting because it connects a practical engineering decision with a simple geometric property of the embedding space. Even if one does not fully accept a causal interpretation of spectral entropy, the empirical result that instruction-induced embedding geometry can predict downstream retrieval quality is useful and worth knowing. The method is simple, inexpensive, and potentially helpful in cold-start settings where labels, corpora, or large evaluation budgets are limited.

**Claims And Evidence:**

Yes

**Claims Explanation:**

Overall, the main claims are supported by fairly broad and clear empirical evidence. The paper evaluates the proposed spectral-entropy-based instruction selection method across a large set of embedding models, retrieval datasets, and candidate instructions, and reports consistent correlations with downstream retrieval quality. The comparisons with retrieval-free baselines, robustness checks across proxy sources, and analyses of failure cases make the empirical story reasonably convincing.

**Requested Changes:**

First, I would encourage the authors to clarify and, if possible, further investigate the causal interpretation of the proposed geometric signal. The current evidence convincingly shows that uncentered spectral entropy is correlated with downstream instruction quality, but it does not fully establish that higher spectral entropy itself causes better retrieval performance. There may be a deeper underlying factor that affects both the embedding geometry and the retrieval outcome. Additional evidence, such as controlled interventions on the embedding geometry, ablations that isolate the effect of spectral spread from other instruction-induced changes, or a more explicit discussion of what can and cannot be interpreted causally, would make the claim stronger and avoid over-interpreting the correlation.

Second, I wonder whether the method could be extended from global instruction selection to more query-adaptive instruction selection. As currently formulated, the paper ranks a pool of candidate instructions and selects a strong default instruction, or a shortlist, for a target retrieval task. However, the best instruction on average across a dataset may not be the best instruction for every type of query. Since queries can be heterogeneous, it would be interesting to cluster the proxy texts or queries into semantic or structural groups, and then apply the same spectral-entropy screening procedure within each cluster. This could test whether different query types benefit from different instructions, and whether cluster-specific instruction selection improves over using a single global instruction. Such an experiment would strengthen the practical story and clarify whether IPP is only a default-instruction selector or could also support more personalized/query-adaptive retrieval systems.

---

> ### Author Response · Authors · 2026-06-27
> **Response to Reviewer SK32**
>
> We thank the reviewer for the positive assessment and for the two constructive suggestions. We agree that the causal interpretation should be stated carefully. Our intended claim is predictive and diagnostic rather than interventional: IPP uses uncentered spectral entropy as a pre-retrieval signal for ranking candidate query-side instructions, not as evidence that artificially increasing entropy would necessarily cause better retrieval. We will revise the abstract/introduction and add an explicit discussion in Sec. 5 to avoid causal overstatement.
>
> The existing analyses already support this narrower interpretation. Proposition 1 only links the uncentered spectrum to pairwise cosine concentration and explicitly does not imply retrieval quality by itself. Empirically, across the full 16×17 benchmark, spectral entropy achieves median oriented Spearman ρ = 0.806 and median regret@1 = 0.004 NDCG@10, showing that the signal is useful for instruction ranking. We also include geometry ablations: centering the embeddings reduces median ρ from 0.681 to 0.286, indicating that the instruction-induced mean direction is part of the signal rather than nuisance variation. The paraphrase audit further shows that changes in Hspec track within-cluster NDCG changes on average, suggesting that the effect is not solely due to coarse lexical differences between instructions. We will make clear that these analyses support a geometric explanation, but do not establish a unique causal mechanism.
>
> We also appreciate the suggestion on query-adaptive instruction selection. Our current paper deliberately studies the earlier cold-start setting: selecting a strong global default or shortlist before corpus access, labels, or stable retrieval evaluation are available. Query-adaptive selection is a natural extension, but it introduces additional choices—how to cluster queries/proxies, how many examples are needed per cluster for stable spectral estimates, and how to route new queries at inference time. We will add a future-work paragraph describing cluster-specific IPP: applying the same spectral-entropy screening within semantic or structural proxy clusters to produce cluster-level instruction rankings or shortlists. This would extend IPP from global default selection toward adaptive instruction routing, while preserving the retrieval-free spirit of the method.
>
> We will incorporate these clarifications to better distinguish correlation, mechanism, and causal claims, and to position query-adaptive IPP as a promising extension beyond the present scope.